# Evolutions of Self-Rated Health and Social Interactions during the COVID-19 Pandemic Affected by Pre-Pandemic Conditions: Evidence from a Four-Wave Survey

**DOI:** 10.3390/ijerph20054594

**Published:** 2023-03-05

**Authors:** Takashi Oshio, Hiromi Kimura, Shingo Nakazawa, Susumu Kuwahara

**Affiliations:** 1Institute of Economic Research, Hitotsubashi University, Tokyo 186-8601, Japan; 2Survey Research Center, Tokyo 103-0027, Japan; 3Japan Cabinet Office, Tokyo 100-8914, Japan

**Keywords:** adaptation, COVID-19, self-rated health, social interaction, state of emergency

## Abstract

The Coronavirus disease 2019 (COVID-19) pandemic has affected individuals’ self-rated health (SRH) and social interactions, but their evolution during the pandemic needs further investigation. The present study addressed this issue using longitudinal data from 13,887 observations of 4177 individuals obtained from a four-wave nationwide, population-based survey conducted between January and February 2019 (before the pandemic) and November 2022. We compared the evolutions of SRH and social interactions during the pandemic between individuals who interacted with others before the pandemic and those who did not. Three noteworthy findings were obtained. First, deterioration in SRH in response to the declared state of emergency was concentrated on individuals with no pre-pandemic interaction with others. Second, SRH generally improved during the pandemic, although the improvement was more remarkable among previously isolated individuals. Third, the pandemic has promoted social interactions among previously isolated individuals and reduced such chances among those who previously interacted with others. These findings underscore the importance of pre-pandemic social interactions as key determinants of responding to pandemic-related shocks.

## 1. Introduction

### 1.1. Research Purpose

The Coronavirus disease 2019 (COVID-19) pandemic has affected individuals’ health and social interactions, but their evolution during the pandemic needs further investigation. It is known that the pandemic and pandemic-related restriction measures had an adverse impact on health and social interactions, but their evolution during the pandemic needs further investigation. Pre-pandemic conditions, especially in terms of social interactions, may affect the response and adaptation of health to pandemic-related shocks, and social interactions may change during the pandemic. 

The present study addressed these issues, focusing on self-rated health (SRH) and interactions with others, using longitudinal data from a four-wave survey that started before the pandemic and continued until November 2022. We also identified the area (prefecture) in which each respondent lived. Hence, we could determine whether each respondent faced the declared state of pandemic-related emergency and controlled for prefecture-level fixed effects in addition to individual-level fixed effects. This enabled us to capture the evolution of SRH and social interactions more precisely than without controlling for these effects.

### 1.2. Literature Review

Studies have found an adverse effect of the Coronavirus disease 2019 (COVID-19) pandemic on mental health, health behavior, and general health outcomes such as SRH [1,2,3,4,5,6]. Notably, heightened risks of social isolation due to pandemic-related restriction measures worsen health outcomes in terms of stress, anxiety, and psychological distress, among others [7,8,9,10,11,12]. These findings are in line with an established view that social isolation has a negative impact on health [13,14,15].

However, there are skeptical views about the negative impact of increased social isolation on mental health during the pandemic. Some studies showed that pandemic-related restriction measures did not cause much social isolation or have any adverse impact on mental health [16], that the effects of those measures, if any, were short-lived [17], and that a majority of individuals were either resilient or recovered quickly from the initial COVID-19 shock [18].

These mixed results may reflect at least two things. First, heterogeneity across individuals in terms of pre-pandemic social interactions may make the observed impact of the pandemic non-uniform. For instance, pre-pandemic social isolation or a lack of social capital is likely to make individuals more sensitive to the onset of the pandemic [19,20]. However, the possibility that pre-pandemic social integration, which was impeded by the pandemic, may add to mental health risks cannot be excluded [21]. Second, the impact of the pandemic is likely to vary over time, although most previous studies have relied on cross-sectional or longitudinal datasets that covered only an initial or earlier stage of the pandemic. Individuals may adapt to any pandemic-related shock over time, along with changes in social behavior. In general, the pace and magnitude of such an adaptation are likely to be affected by an individual’s before-the-shock conditions [22].

### 1.3. Hypotheses

Based on these observations from previous studies and their limitations, we examined how SRH and social interactions changed for approximately 20 months after the pandemic outbreak. While the pandemic outbreak is expected to adversely affect SRH, we hypothesized that the impact was the most serious among socially isolated individuals before the pandemic. We further hypothesized that shock adaptation would be affected by social interactions before the pandemic, although it is difficult to predict the direction and pace of the adaptation.

## 2. Materials and Methods

### 2.1. Study Samples

We used data obtained from population-based, nationwide Internet surveys conducted in January and February 2019 and February 2020 (Wave 1; before the pandemic), March 2021 (Wave 2), October/November 2021 (Wave 3), and November 2022 (Wave 4). We distributed questionnaires to registrants of an Internet survey company.

To construct the pre-pandemic benchmark sample in Wave 1, we planned to collect data from approximately 5000 participants: around two-thirds from the survey between 25 January and 7 February 2019, and the remaining one-third from the survey between 7 February and 20 February 2020. The target samples were then divided into two groups. First, approximately 3800 registrants were distributed equally between each of the 47 prefectures, between men and women, and among the five age groups (15–24, 25–34, 35–44, 45–59, and 60+ years of age). Next, we allocated approximately 1400 registrants to each gender-age group in each prefecture in proportion to each prefecture’s actual population size. Therefore, the sample was not representative of the Japanese population. With a target of collecting data from approximately 5000 participants, we distributed questionnaires to registrants of an Internet survey company and made questionnaires available to the registrants during the survey period. Data from 4177 participants were obtained when the survey was closed.

We conducted a Wave 2 survey between 3 March and 11 March 2021, when Japan faced the third wave of COVID-19 infection. During the survey, four prefectures in the Tokyo metropolitan area (Tokyo, Chiba, Saitama, and Kanagawa) were still in a state of emergency; immediately after, six prefectures (Aichi, Gifu, Osaka, Hyogo, Kyoto, and Fukuoka) lifted the state of emergency on 28 February 2021. We sent the questionnaire to the pooled survey monitors who participated in Wave 1 and collected data from 2260 individuals.

We conducted a Wave 3 survey between 28 October and 8 November 2021, a month after all prefectures had lifted their state of emergency on 30 September 2021. We sent the questionnaire to those who participated in the Waves 1 and 2 surveys and collected data from 2260 individuals.

Finally, we conducted a Wave 4 survey between 2 November and 21 November 2022, when the number of new COVID-19 infections was in the trough between the seventh and eighth waves. We sent the questionnaire to those who participated in the survey in all three previous waves and collected data from 1834 individuals.

Figure 1 summarizes how we constructed four-wave data. For the statistical analysis, we used unbalanced longitudinal data from 13,883 observations of 4177 individuals. A full set of the survey questionnaire is available from Appendix A.

### 2.2. Self-Rated Health

As a key health outcome, we focused on SRH, which represents overall health conditions [23,24]. The survey asked participants to answer the question “How do you feel about your health condition?” by selecting good, somewhat good, average, somewhat poor, and poor, respectively. We constructed a binary variable for poor SRH by allocating one to those who answered poor or somewhat poor and zero to others, considering that the score is generally skewed toward better health.

### 2.3. Social Interactions

Regarding social interactions, we considered interactions with others—such as friends, colleagues, and neighbors, and excluding family members—by focusing on the respondent’s response to the question, “How often are you interacting (e.g., meeting and communicating) with others on average?” We constructed a binary variable of social isolation by allocating one to those who answered that they did not interact with anyone and zero to others. We also considered two aspects of social interaction: receiving social support and using social networking services (SNS). We constructed a binary variable for receiving no social support by allocating one to those who answered yes to the question, “Do you have any family members or friends who support you if you are in trouble?” and zero to others. Regarding SNS use, we constructed a binary variable for SNS use by allocating one to those who answered almost every day, three to four times a week, or once a week to the question, “How often do you use SNS (such as Facebook, Twitter, Line, and Instagram) in these days?” and zero for those who answered less frequently or never, respectively.

### 2.4. Individual-Level Covariates

As covariates, we considered marital status (married or single), occupational status (regularly employed, non-regularly employed, self-employed, unemployed, not working, or student), educational attainment (junior high school, high school, junior college, or college or above), and age (20 s or below, 30 s, 40 s, 50 s, 60 s, or above). We constructed binary variables for each category of variables. We also construct binary variables for each household income quartile.

### 2.5. Analytic Strategy

We divided respondents into two groups: those who were socially isolated before the pandemic (hereafter referred to as SIBP)—that is, they had no interaction with others in Wave 1—and others (No-SIBP). We also divided 47 prefectures into two areas: the area consisting of 10 prefectures where the pandemic-related state of emergency was declared in Wave 2 or lifted just before that wave—that is, Tokyo, Chiba, Saitama, Kanagawa, Aichi, Gifu, Osaka, Hyogo, Kyoto, and Fukuoka—and the area consisting of the remaining 37 prefectures with no declaration of a state of emergency over four waves. Hereafter, we refer to these areas as ES and No-ES areas, respectively.

For descriptive analysis, we compared the evolution of poor SRH, interaction with others, receiving social support, and SNS use over four waves in each of the ES and No-ES areas between SIBP and No-SIBP without controlling for any other variable. For the regression analysis of the evolution of poor SRH, we estimated the following linear probability model (LPM), which controls for individual- and prefecture-level fixed effects [25,26], to explain it for individual *i* in prefecture *p* in Wave *t*:(1)$$Poor\;SRH_{ipt}=\sum_{t=2}^{4}\beta_{t}Wave_{t}+\sum_{t=2}^{4}\gamma_{t}Wave_{t}\times SIBP_{i}+{\left(covariates\right)}_{it}+u_{i}+v_{p}+\varepsilon_{ipt}$$
where *Wave* indicates a binary variable for each wave; *u* and *v* are individual- and prefecture-level fixed effects, respectively; and *ε* is an error term.

The estimated value of *β_t_* indicates a change in the probability of poor SRH from Wave 1 to Wave *t* for No-SIBP individuals. The estimated value of *β_t_ + γ_t_*, which is calculated after regression, indicates a change in the probability of poor SRH from Wave 1 to Wave *t* for SIBP individuals.

Although the dependent variable is binary, we estimated an LPM rather than a probit or logistic model because an LPM enables us to interpret the estimated coefficient of the interaction term additively. We estimated the abovementioned equation separately for each ES and No-ES area. We also replaced poor SRH with binary variables of interaction with others, receiving social support, and SNS use.

To mitigate the attrition bias, we applied inverse probability weighting [27]; we first used the probit model to predict the probability of staying in the survey until Wave 4 for each participant using their attributes observed in Wave 1. We then used the inverse of the predicted probability as a weight in our regression analysis.

For robustness, we repeated the same regression analysis by removing 411 respondents infected with COVID-19 from the sample. The Stata software package (version 17) was used for all statistical analyses.

## 3. Results

### 3.1. Descriptive Analysis

Table 1 summarizes the key features of the participants at baseline (Wave 1, before the pandemic). Of the 4117 participants, 558 (14.1%) had no interaction with others; they were categorized as having SIBP. No interaction with others was positively associated with being male and single, having an unstable occupational status (non-regularly employed and unemployed), low educational attainment, and low income. Out of all participants, 22.9% lived in areas where the state of pandemic-related emergency was declared in Wave 2 (ES areas).

Table 2 summarizes the prevalence of poor SRH and the three types of social interactions, and their correlations in terms of pairwise correlation coefficients for all respondents (individuals × waves). Poor SRH was negatively associated with interaction with others, receiving social support, and SNS use; social interaction was positively associated with each other.

Table 3 shows the evolution of interaction with others and each social behavior during the pandemic without controlling for any other variable. The right part of the table shows that the probability of poor SRH increased by 9.7 percentage points in response to the state of emergency declared in Wave 2 among SIBP individuals; all other combinations of pre-pandemic social interactions and residential areas demonstrated relatively limited responses in that wave. Meanwhile, in all cases, the probability of poor SRH gradually declined—even below the pre-pandemic level—over subsequent waves, and individuals with SIBP exhibited the most remarkable improvement.

### 3.2. Regression Analysis

After controlling for individual- and prefecture-level fixed effects as well as covariates, Table 4 presents the estimated values of *β*_2_, *β*_3_, and *β*_4_ (in terms of percentage points), which indicate the changes in each of the four dependent variables from Wave 1 to each wave among No-SIBP individuals. The table also reports the estimated values of *β*_2_ + *γ*_2_, *β*_3_ + *γ*_3_, and *β*_4_ + *γ*_4_, which indicate those for individuals with SIBP. Based on the results in Table 4, Figure 2 and Figure 3 graphically illustrate the evolution of poor SRH and its interaction with others, respectively. The detailed regression results are provided in Appendix A.

As seen in the first set of columns in Table 4 and Figure 2, the jump in the probability of poor SRH in Wave 2 was concentrated on SIBP individuals living in ES areas, with a 9.8 percentage point (95% confidence interval [CI]: 0.1–19.5) increase. In Wave 3, those individuals experienced a remarkable reduction in the probability of poor SRH to a level that was 8.9 percentage points (95% CI: 1.4–16.4) below the pre-pandemic level. The probability of poor SRH declined among all types of individuals over time and fell below the pre-pandemic level, while adaptation was more substantial among SIBP individuals.

The second set of columns in Table 4 and Figure 3 indicate a sharp contrast in the evolution of the interaction between SIBP and No-SIBP individuals. More than 20% of SIBP individuals, whether living in ES or No-ES areas, started to interact with others in Wave 2 and maintained social interaction in subsequent waves, while No-SIBP individuals slightly reduced interaction with others during the pandemic.

Behind this evolution of interaction with others, the third set of columns shows that the probability of receiving social support rose by 1.9–8.6 percentage points in Waves 2–4 among SIBP individuals, while the probability declined by 0.1–4.1 percentage points over the same period among No-SIBP individuals. SNS use exhibited more mixed evolutions, but it is remarkable that the probability of SNS use jumped by 19.4 (95% CI: 8.9–29.9) percentage points in Wave 2 among SIBP individuals living in ES areas.

The regression analysis results in Table 4 are generally consistent with those of the descriptive analysis in Table 3. Appendix A summarizes the key regression results obtained after excluding respondents infected with COVID-19. The magnitude, sign, and statistical significance of each regression coefficient are similar to those observed in Table 4.

## 4. Discussion

This study examined how SRH and social interactions evolved during the COVID-19 pandemic, focusing on the impact of pre-pandemic social interactions. We took full advantage of a four-wave survey starting before the pandemic outbreak and covering the period ending in November 2022. Unlike in most previous studies, we further controlled for individual- and area (prefecture)-level fixed effects.

We obtained three noteworthy findings. First, deterioration in SRH was concentrated on SIBP individuals who faced a state of emergency in the ES areas. The impact of the pandemic on SRH was relatively limited among No-SIBP individuals, even when they lived in ES areas. This finding suggests that social isolation made individuals psychosocially fragile to pandemic-related restrictions; in other words, the pre-pandemic interaction with others contributed to an individual’s resilience to the shock.

Second, the probability of poor SRH generally decreased during the pandemic, although its pace differed between SIBP and No-SIBP individuals and between ES and No-ES areas. In this regard, two additional findings should be noted. First, in Wave 3, the probability of poor SRH more than fully reversed the jump in Wave 2 among SIBP individuals. This indicates that the impact of the restriction measures was short-lived, as suggested by some studies [17,18]. Second, the probability of poor SRH fell below the pre-pandemic level by November 2022 among all types of individuals. The feeling of escaping from the worst time, which was accompanied by mobility restrictions, made individuals feel even healthier than before the pandemic. This evolution implied a shift of the “reference point” for subjectively evaluating one’s health or more general well-being [28]. This shift seems to have been the greatest among SIBP individuals living in ES areas in Wave 3; they experienced substantial improvement in SRH, a seeming payback triggered by the lifted state of emergency.

Third, the pandemic affected social interactions differently between SIBP and No-SIBP individuals. The pandemic promoted interaction with others among SIBP individuals and reduced the chances of doing so among No-SIBP individuals. The latter response was consistent with observations in previous studies, which argued that mobility restrictions impeded social interactions [7,8,9,10,11,12]. The former response suggests that pandemic-related shocks urged SIBP individuals, especially during an emergency, to free themselves from isolation by constructing or strengthening interactions with others. This interpretation is consistent with coping theory [29], which argues that individuals attempt to deal with specific demands of a situation that burdens or exceeds their resources. This coping effort was underscored by the observation that SIBP individuals received more social support during the pandemic and that the declared state of emergency triggered SNS use in the ES areas. The need to strengthen the interaction with others was not so imminent among No-SIBP individuals, who could interact with others or at least felt connected. Encouraged social interactions among SIBP individuals probably accounted for their smoother adaptation to SRH during the pandemic than among No-SIBP individuals.

Overall, we observed that the impacts of COVID-19 on SRH and social interactions were mixed and varied over time. We also found that pre-pandemic social interaction affected both sensitivities to its shock and adaptation.

However, we acknowledge several limitations of this study, in addition to the potential biases inherent in an Internet survey and attribution biases. First, our analysis was based on the answers to simple questions regarding SRH, social interaction, receiving social support, and SNS use, which ignored various aspects of each domain. Second, we did not examine the two-way causation between SRH and social interactions; our analysis was limited to their evolution at the onset of the pandemic and over the subsequent period. As seen in Table 2, SRH and interaction with others were closely associated, and we could predict an interaction between health and social interactions. Third, the impact of the declared state of emergency is likely to have spilled over to neighboring prefectures [30], suggesting that our dichotomy of ES and No-ES areas may be misleading. Fourth, the adaptation and evolution patterns of SRH, social interactions, and their interaction are likely to change further if COVID-19 infections linger, requiring caution in any generalization of the obtained results. More longitudinal data are required to obtain a more in-depth understanding of their dynamics. Lastly, we should be cautious in generalizing the findings of this study based on the Japanese survey data. The magnitudes of pandemic-related shocks and restriction measures as well as sociocultural contexts related to social interactions may differ from country to country.

## 5. Conclusions

Within the limitations of this study, the results underscore the importance of pre-pandemic social interactions as a key determinant of the evolution of SRH and social interactions during the COVID-19 pandemic. We conclude that pre-pandemic social isolation amplified the negative shock from the pandemic but accelerated the adaptation to it. These findings suggest that policy measures are required to prevent social isolation to mitigate external health shocks in advance and hence increase public health resilience. While we can expect socially isolated individuals to adapt to shocks over time, policy support to encourage them to interact with others is recommended for successful adaptation.

## Figures and Tables

**Figure 1 ijerph-20-04594-f001:**
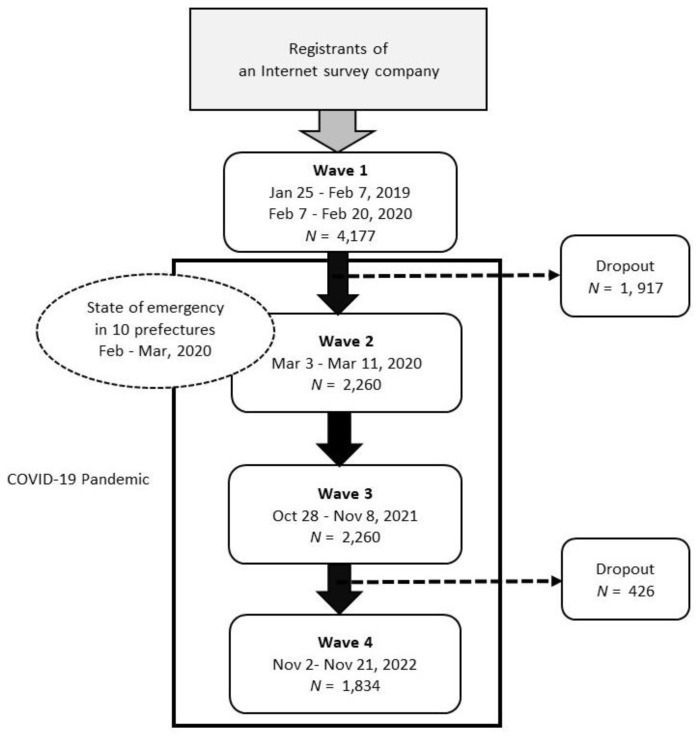
Flowchart of subjects included for the analysis.

**Figure 2 ijerph-20-04594-f002:**
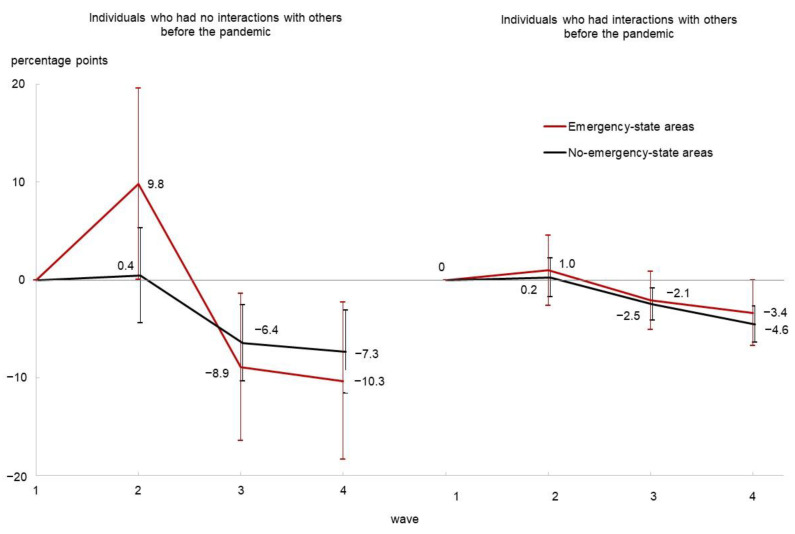
The change in the probability of poor self-rated health from Wave 1 (before the pandemic). Note. Error bars indicate 95% confidence intervals.

**Figure 3 ijerph-20-04594-f003:**
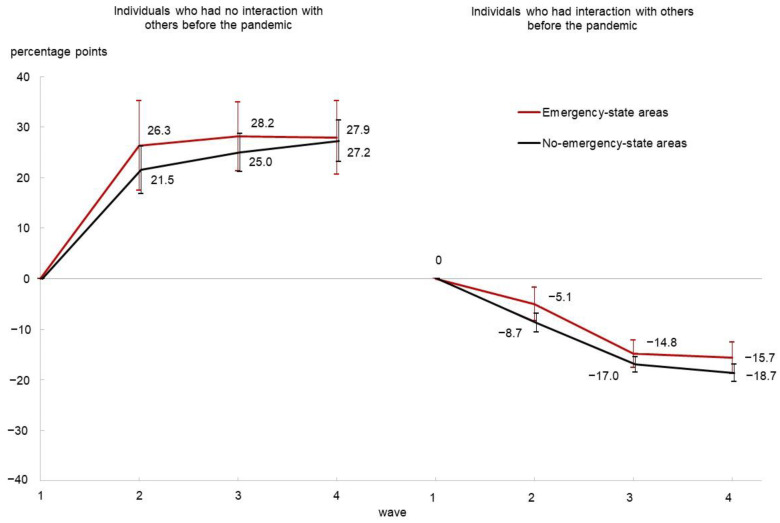
The change in the probability of interaction with others from Wave 1 (before the pandemic). Note. Error bars indicate 95% confidence intervals.

**Table 1 ijerph-20-04594-t001:** Key features of the participants at baseline (Wave 1; before the pandemic).

Interaction with Others		Interaction	No Interaction	All
Proportion (%)				
Sex				
	Men	46.9	63.3	49.2
	Women	53.1	36.7	50.8
Occupational status				
	Regularly employed	41.6	34.9	40.7
	Non-regularly employed	22.7	27.7	23.4
	Self-employed	7.5	7.8	7.6
	Unemployed	2.9	7.0	3.5
	Not working	20.9	21.1	20.9
	Student	4.4	1.5	4.0
Educational attainment				
	Junior high school	1.8	6.0	2.4
	High school	40.9	52.4	42.5
	Junior college	13.1	8.3	12.4
	College or above	44.2	33.3	42.6
Married		59.8	42.9	57.5
Lived in emergency-state areas		23.1	21.6	22.9
Age (years)				
	*M*	48.5	48.5	48.4
	*SD*	(15.6)	(15.2)	(15.4)
Household income (annual, million JPY)				
	*M*	6.21	4.92	6.03
	*SD*	(6.64)	(7.73)	(6.82)
*N*		3589 (85.9%)	588 (14.1%)	4177 (100%)

**Table 2 ijerph-20-04594-t002:** Poor self-rated health and social interactions: Prevalence and correlations (*N* = 13,887).

	Prevalence (%)	Pairwise Correlation Coefficient		
		Poor self-Rated Health	Interaction with Others	Receiving Social Support
Poor self-rated health	20.4			
Interaction with others	79.2	–0.094 ***		
Receiving social support	88.0	–0.120 ***	0.306 ***	
SNS use	66.5	–0.026 **	0.163 ***	0.107 ***

*** *p* < 0.001, ** *p* < 0.01.

**Table 3 ijerph-20-04594-t003:** The evolution of self-rated health and social behavior during the pandemic (*N* = 13,887).

	Probability (%)				Change from Wave 1 (Percentage Points)			
Interaction with others in Wave 1	No		Yes		No		Yes	
State of emergency in Wave 2	Yes	No	Yes	No	Yes	No	Yes	No
Poor self-rated health								
Wave 1	18.9	20.9	37.0	34.9				
Wave 2	19.7	20.1	46.7	32.9	0.8	−0.8	9.7	−2.0
Wave 3	17.5	18.4	29.1	28.4	−1.4	−2.5	−7.9	−6.5
Wave 4	16.1	15.9	27.8	27.5	−2.8	−5.1	−9.2	−7.5
Interaction with others								
Wave 1	100.0	100.0	0.0	0.0				
Wave 2	94.3	90.9	23.3	20.9	−5.7	−9.1	23.3	20.9
Wave 3	84.5	83.0	26.8	24.9	−15.5	−17.0	26.8	24.9
Wave 4	84.0	81.0	26.9	27.2	−16.0	−19.0	26.9	27.2
Receiving social support								
Wave 1	95.1	93.1	59.8	61.8				
Wave 2	93.4	92.1	66.7	69.9	−1.7	−0.9	6.8	8.1
Wave 3	91.2	90.9	68.5	63.8	−3.9	−2.1	8.7	2.0
Wave 4	91.7	90.1	63.0	66.4	−3.4	−3.0	3.1	4.6
SNS use								
Wave 1	69.8	67.6	44.1	47.3				
Wave 2	74.4	72.3	70.0	49.8	4.6	4.7	25.9	2.5
Wave 3	69.4	70.8	49.6	49.2	−0.4	3.1	5.5	2.0
Wave 4	68.4	67.3	51.9	46.7	−1.4	−0.3	7.8	−0.6

**Table 4 ijerph-20-04594-t004:** Key estimation results of regression models to explain the probabilities of poor self-rated health and social interactions.

	Poor Self-Rated Health		Interaction with Others		Receiving Social Support		SNS Use	
	Coef. × 100	95% CI ^1^	Coef. × 100	95% CI	Coef. × 100	95% CI	Coef. × 100	95% CI
Emergency-state areas (*N* = 3146)								
*β* _2_	1.0	(−2.6, 4.5)	−5.1	(−8.3, −1.8)	−1.3	(−4.2, 1.6)	1.4	(−2.5, 5.2)
*β* _3_	−2.1	(−5.1, 0.9)	−14.8	(−17.6, −12.1)	−4.1	(−6.5, −1.7)	−0.2	(−3.4, 3.0)
*β* _4_	−3.4	(−6.7, 0.0)	−15.7	(−18.7, −12.6)	−4.1	(−6.8, −1.4)	−0.3	(−3.9, 3.3)
*β*_2_ + *γ*_2_	9.8	(0.1, 19.5)	26.3	(17.4, 35.2)	8.0	(0.1, 16.0)	19.4	(8.9, 29.9)
*β*_3_ + *γ*_3_	−8.9	(−16.4, −1.4)	28.2	(21.3, 35.0)	8.6	(2.5, 14.6)	5.1	(−3.0, 13.1)
*β*_4_ + *γ*_4_	−10.3	(−18.3, −2.3)	27.9	(20.6, 35.3)	2.0	(−4.5, 8.5)	9.8	(1.1, 18.4)
No-emergency-state areas (*N* = 10,727)								
*β* _2_	0.2	(−1.7, 2.2)	−8.7	(−10.6, −6.8)	−0.1	(−1.7, 1.5)	2.5	(0.4, 4.6)
*β* _3_	−2.5	(−4.1, −0.9)	−17.0	(−18.6, −15.4)	−2.1	(−3.4, −0.8)	3.3	(1.6, 5.0)
*β* _4_	−4.6	(−6.4, −2.7)	−18.7	(−20.4, −16.9)	−2.5	(−4.0, −1.0)	1.8	(−0.1, 3.8)
*β*_2_ + *γ*_2_	0.4	(−4.4, 5.3)	21.5	(16.9, 26.2)	7.2	(3.2, 11.2)	−1.4	(−6.5, 3.8)
*β*_3_ + *γ*_3_	−6.4	(−10.3, −2.5)	25.0	(21.2, 28.7)	1.9	(−1.3, 5.1)	1.9	(−2.2, 6.1)
*β*_4_ + *γ*_4_	−7.3	(−11.5, −3.1)	27.2	(23.2, 31.3)	3.7	(0.2, 7.1)	0.8	(−3.7, 5.3)

^1^ Confidence interval.

## Data Availability

Data supporting the reported results are available from https://www5.cao.go.jp/keizai2/wellbeing/manzoku/index.html (accessed on 13 February 2023).

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
