# Peer review of "Evolutions of Self-Rated Health and Social Interactions during the COVID-19 Pandemic Affected by Pre-Pandemic Conditions: Evidence from a Four-Wave Survey"

_ijerph, 2023, doi:10.3390/ijerph20054594_

Round 1

Reviewer 1 Report

this is an interesting study using several dataset. and it can be accepted after minor revision.

1. please redo your table and table title position.

2. please create a sample flow chart from wave 1 until 4 to have better understanding about the number of individuals in and out of survey

3. another limitation for your study is that your questionnaire is too simple with only one question for each domain, SRH, social interaction, and SNS use. with only one question we can't get a detailed answer for SRH, social interaction, and SNS use.

Reviewer 2 Report

This is a very important study with strong relevance. However, the manuscript has the following problems that require further improvement and refinement:

1.In the introductory section, the theoretical contribution of this study is not sufficiently elaborated and needs to be further strengthened.

2.After the introduction, there should be a section on literature review and hypothesis development, which the authors need to add.

3.In the sample section of the study, there is insufficient description of the sampling method for the first round of the survey and how to ensure that the sample is representative. These elements need to be added on.

4.The absence of content on survey methodology and quality control of the survey makes the quality and credibility of the data questionable. The authors need to add these elements.

5.There is a large difference in the number of samples from the first round of the survey to the fourth round of the survey. In this case, how to ensure that the samples from all four rounds of the survey are equally representative needs to be explained by the authors.

6.English abbreviations are required in full the first time they appear in a manuscript.

7.No distinction was made between the objects of interaction in terms of social interaction variables, which is a huge limitation of the study. Interactions with family members and interactions with strangers are two types of social interactions that are not completed and, therefore, need to be distinguished.

8.In the discussion section, the issue of external validity of the study needs to be explored, i.e., whether the relevant findings can be applied outside of Japan.

9.The conclusions are too simple and need further development.

Round 2

Reviewer 2 Report

It can be published.